# TokenMixup: Efficient Attention-guided Token-level Data Augmentation for Transformers

**Hyeong Kyu Choi**[*], **Joonmyung Choi**[*], **Hyunwoo J. Kim**[†]
Department of Computer Science and Engineering, Korea University
{imhgchoi, pizard, hyunwoojkim}@korea.ac.kr

## Abstract

Mixup is a commonly adopted data augmentation technique for image classification. Recent advances in mixup methods primarily focus on mixing based on saliency. However, many saliency detectors require intense computation and are especially burdensome for parameter-heavy transformer models. To this end, we propose TokenMixup, an efficient attention-guided token-level data augmentation method that aims to maximize the saliency of a mixed set of tokens. TokenMixup provides ×15 faster saliency-aware data augmentation compared to gradient-based methods. Moreover, we introduce a variant of TokenMixup which mixes tokens within a single instance, thereby enabling multi-scale feature augmentation. Experiments show that our methods significantly improve the baseline models' performance on CIFAR and ImageNet-1K, while being more efficient than previous methods. We also reach state-of-the-art performance on CIFAR-100 among from-scratch transformer models. Code is available at https://github.com/mlvlab/TokenMixup.

## 1 Introduction

Various data augmentation methods have been proposed for computer vision tasks. One of the most successful augmentation methods is mixup [1], which is commonly applied to image classification. It attempts to augment the input by taking a convex combination of two random instances, and by reassigning the ground truth label correspondingly. Building on mixup, many subsequent works [2–9] have been introduced. They focus on mixing instances in a more meaningful way (*e.g.*, saliency-aware mixup, manifold level mixup, submodular diversity maximization *etc.*), while a great majority attempts to perform mixup based on saliency.

These saliency-aware mixup methods often entail gradient computation. To detect salient regions, the input is forward-propagated once and back-propagated to extract the gradient map. However, such a mechanism tends to be computationally heavy. This is especially pertinent to transformer-based models [10], as they generally retain high parameter volume. Keeping this in mind, we aim to present a saliency-aware mixup method favorable for transformers. Specifically, we regard self-attention as the inherent saliency detector, which serves as an efficient means for augmentation.

We accordingly propose **TokenMixup**, an efficient attention-guided token-level mixup method. Its objective is to mix intermediate token sets so that the saliency level of a batch is maximized. To detect salient tokens, we take advantage of the attention map to achieve a ×15 speed-up compared to the gradient-based saliency estimator, without compromising model performance. The batch saliency is maximized by optimally matching instance pairs, accomplished by an algorithm that exactly solves the optimization problem. We also introduce ScoreNet, a simple module that measures the difficulty of an instance. Based on ScoreNet, TokenMixup is applied selectively with respect to sample difficulty, which can be viewed as a type of curriculum learning method. To the best of

---

[*]The first two authors contributed equally
[†]corresponding author

our knowledge, this is the first attempt in the literature to discuss mixup from a curriculum learning perspective. Furthermore, we present a novel mixup approach that combines intermediate tokens across transformer layers within a single instance. We name this Vertical TokenMixup (VTM) since it *vertically* mixes tokens to provide rich multi-scale information to the attention layer.

Experiments on image classification datasets imply that our methods are both effective and efficient. Also, an advantage of TokenMixup is that it be used in parallel with other mixup methods. In combination with those methods, we improve our baselines' performance by significant margins on CIFAR-10, CIFAR-100, and ImageNet-1K. Especially for CIFAR-100, we achieve a new state-of-the-art performance among transformer-based models that do not use pre-trained models.

Then, our contributions are fourfold:

- We propose TokenMixup, an efficient attention-guided token-level mixup for transformers. By incorporating the self-attention map, we achieve ×15 faster saliency-aware data augmentation compared to gradient-based methods.

- We incorporate curriculum learning into mixup, to adaptively perform TokenMixup in an intermediate layer based on the confidence score of ScoreNet.

- We also studied a variant, Vertical TokenMixup, which performs mixup with a single sample and enables multi-scale feature augmentation.

- We achieve state-of-the-art performance on CIFAR-100 among transformer models trained from scratch. We also gain significant improvements over baseline models for image classification.

## 2 Related Works

**Mixup.** Input Mixup [1] is a data augmentation method widely adopted for image classification. The classification model is trained with a convex combination of input images and labels. A special case of Input Mixup is CutMix [2], which can be seen as a pixel-wise mixup method with binary masks. Recent advances in these mixup methods mainly focus on appropriately utilizing saliency information, and on mixing in image feature levels.
The core motivation of saliency-based mixup is that salient regions should be preserved when mixed, to retain a sufficient amount of information and learn more natural feature representations. SaliencyMix [5] adopts various saliency detectors to directly extract salient regions. Puzzle Mix [3] attempts to mix images using saliency information while retaining local statistics. Co-Mixup [4] maximizes gradient-based saliency while encouraging supermodular diversity of the mixed images. SuperMix [9] takes advantage of supervised signals to mix input images based on saliency.
On the other hand, Manifold Mixup [6] has provided theoretical grounds on the advantages of mixing images in the higher level of features. Manifold mixing helps the model learn flatter representation and smoother decision boundaries. Moreover, it is noted that it enhances robustness to adversarial attacks. Recently, AlignMix [7] was proposed to further align geometric properties of objects by mixing instances in the feature space. Furthermore, StyleMix [8] proposed to manipulate content and style information disparately to generate more abundant and robust samples.

**Vision Transformers.** Originating from the natural language processing field, the Transformer [10] model enjoyed tremendous success in numerous computer vision applications, *e.g.*, image classification, object detection [11, 12], and human-object interaction detection [13, 14] *et cetera*. Specifically, image classification is one of the most fundamental computer vision tasks. Starting from the ViT [15], many works have attempted to assimilate the convolutional operation with transformer blocks [16–21]. Other variants have effectively incorporated the convolutional neural network into the transformer to achieve superior performance [22–26].

## 3 Methods

Here, we introduce TokenMixup, a simple but effective mixup method for transformer models. The goal of our method is to augment intermediate tokens while maximizing the saliency level. This can be accomplished in three major steps: 1) Sample difficulty assessment, 2) Attention-guided saliency detection & optimal assignment, 3) Token-level mixup. The following subsections provide detailed descriptions of the steps.

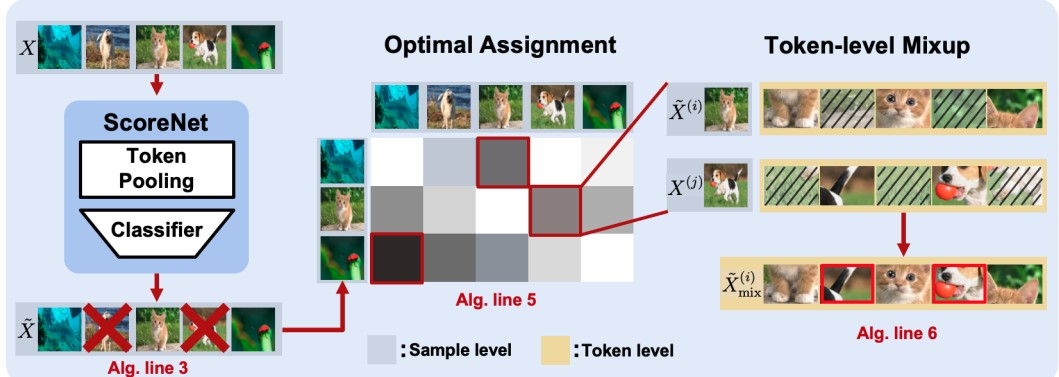

Figure 1: **TokenMixup.** Batch samples $X$ are first filtered with respect to difficulty scores evaluated by ScoreNet in $\mathcal{F}$, resulting in *easy* samples $\tilde{X}$. Then, images in $\tilde{X}$ are optimally paired with samples in $X$ via Hungarian matching, so that the overall saliency of the mini-batch is maximized after mixup. For a matched pair $(\tilde{X}^{(i)}, X^{(j)})$, tokens of $\tilde{X}^{(i)}$ with sufficiently lower saliency levels than the tokens of $X^{(j)}$ are replaced, resulting in a new token set $\tilde{X}^{(i)}_{\text{mix}}$ containing a greater saliency level.

## 3.1 Sample Difficulty Assessment

We first assess the difficulty of the input to adaptively decide whether to apply augmentation. To evaluate sample difficulty, a measuring function $\mathcal{F}$ needs to be defined. We utilize a parameterized module, ScoreNet, which is a simple MLP that predicts the target value $Y^{(i)}$ based on the intermediate tokens $X^{(i)}$, where $i$ is the index of the sample in a batch (See the supplement for figure). Given the ScoreNet output, the difficulty score is evaluated with the prediction loss computed as

$$\mathcal{F}(X^{(i)}, Y^{(i)}) = \text{CrossEntropy}(\text{ScoreNet}(X^{(i)}), Y^{(i)}). \tag{1}$$

If the score is greater than a threshold $\tau$, the sample is deemed sufficiently hard and no mixup is performed. On the other hand, the set of *easy* samples with $\mathcal{F}(X^{(i)}, Y^{(i)}) < \tau$ denoted as $\tilde{X}$ will be augmented by Token-level mixup after pairing with other samples in a mini-batch, as illustrated in Figure 1. Note that ScoreNet is trained simultaneously with the main model and training ScoreNet to assess sample difficulty at the intermediate layer can be viewed as an auxiliary task.

## 3.2 Attention-guided Saliency Detection & Optimal Assignment

Several works [19, 27–29] have discussed the imbalanced information of tokens. Due to this imbalance, random token mixing would potentially cause significant information loss and meaningless token replacements (See the supplement for relevant analysis). Thus, we aim to mix tokens based on saliency.

**Attention-guided saliency detection.** Instead of the computationally heavy gradient-based saliency detectors [3, 4], we take advantage of the attention map, an inherent saliency approximator within transformer modules. Specifically, we can infer the saliency of the tokens in the $i^{th}$ layer via Attention Rollout [30], which computes the attention imposed on the tokens from layer $i$ to $i + \ell$ ($\ell \geq 0$) as

$$A = \Phi^{(i)} \cdot \Phi^{(i+1)} \cdots \Phi^{(i+\ell)}, \tag{2}$$

such that $\Phi^{(i)} = \frac{1}{H} \sum_{h=1}^{H} \Phi_h^{(i)}$ where $H$ is the number of heads in the multi-head attention layer, and $\Phi_h^{(i)}$ is the $h^{th}$ attention head of layer $i$. In order to retrieve $A$, the samples need to be propagated for the subsequent $\ell$ layers with stop-gradient. Since each additional forward propagation is an overhead, we approximate Attention Rollout by setting $\ell = 0$. That is, we only use the attention map of the following layer as the saliency estimator. We find this a sufficient approximation of the full Attention Rollout. See section 5.1 for analysis on the soundness of the approximated saliency map. Then, the saliency score $S_t$ can be computed as

$$S_t = \frac{1}{n} \sum_{i=1}^{n} A_{i,t}, \tag{3}$$

where $t = 1, 2, \ldots, n$ denotes the token index.

---
**Algorithm 1** TokenMixup
---
**Input:** $X \in \mathbb{R}^{b \times n \times d}$, $Y \in \mathbb{R}^{b \times c}$, $\tau$, $\rho$
**Output:** $X \in \mathbb{R}^{b \times n \times d}$, $Y \in \mathbb{R}^{b \times c}$

1:  $U^{(i)} \leftarrow$ evaluate difficulty of $X^{(i)}$    s.t. $i = 1, 2, \ldots, b$                          ▷ Eq. (1)
2:  $S_t^{(i)} \leftarrow$ evaluate saliency of each token with $\ell$-step attention rollout           ▷ Eq. (3)
3:  $\tilde{X}^{(i)}, \tilde{Y}^{(i)}, \tilde{S}^{(i)} \leftarrow$ select easy samples w.r.t $U$ and $\tau$    s.t. $i = 1, 2, \ldots, b'$
4:  $C_{ij} \leftarrow \sum_t \max(S_t^{(j)} - \tilde{S}_t^{(i)} - \rho, 0)$   s.t. $i = 1, 2, \ldots, b'$ and $j = 1, 2, \ldots, b$      ▷ Eq. (6)
5:  $\sigma(m) \leftarrow$ HungarianMatching($C_{ij}$)    s.t. $m = 1, 2, \ldots, b'$ and $\sigma(m) \in \{1, 2, \ldots, b\}$
6:  $\tilde{X}_{\text{mix}}^{(i)} \leftarrow$ Mix-token($\tilde{X}^{(i)}, X^{(\sigma(i))}; \rho$)    s.t. $i = 1, 2, \ldots, b'$                  ▷ Eq. (10)
7:  $\tilde{Y}_{\text{mix}}^{(i)} \leftarrow$ Relabel($\tilde{Y}^{(i)}, Y^{(\sigma(i))}$)                                        ▷ Eq. (11)
8:  restore $\tilde{X}_{\text{mix}}^{(i)}, \tilde{Y}_{\text{mix}}^{(i)}$ to $X, Y$    s.t. $i = 1, 2, \ldots, b'$
9:  **return** $X, Y$
---

**Optimal assignment.**   Based on the estimated saliency of tokens, we aim to maximize the overall saliency level by optimally assigning a different mixup target for each instance. So, we first define $P_{i,j} \in \mathbb{R}^n$ as the saliency difference between a random instance pair $(i, j)$, which is computed as

$$P_{i,j} \overset{\triangle}{=} S_j - S_i, \tag{4}$$

where $i, j = 1, 2, \ldots, b$ and $S_i \in \mathbb{R}^n$ is the token saliency map from the $i^{th}$ instance. Then, we represent our objective as an optimization problem written as

$$\max_{\sigma \in \mathcal{M}} \max_{r_i \in \mathbb{B}^n} \sum_{i=1}^{b} (P_{i,\sigma(i)} - \rho)^{\top} r_i, \tag{5}$$

where $\mathcal{M}$ is the set of all possible batch permutations of the $b$ instances, and $\sigma \in \mathcal{M}$ refers to an arbitrary permutation. $\rho$ is the threshold hyperparameter that controls the minimum saliency gain required for a token to be mixed, and $r_i$ is a binary decision vector for the $i^{th}$ instance. *i.e.*, token $t$ of $i$ is replaced with token $t$ of $\sigma(i)$ when $r_{i,t} = 1$, and preserved when $r_{i,t} = 0$ $(t = 1, 2, \ldots, n)$.

The above optimization problem can be exactly solved by utilizing the Hungarian Matching algorithm [31]. The matching algorithm requires the score matrix, $C \in \mathbb{R}^{b \times b}$, which is computed as

$$C_{i,j} = \sum_{t=1}^{n} \max(P_{i,j}^{(t)} - \rho, 0), \tag{6}$$

where each item in $C_{i,j}$ refers to the maximum saliency gain resulting from mixing $\tilde{X}^{(i)}$ and $X^{(j)}$. The actual maximum gain will be reached when $\rho = 0$, but we set a positive threshold to control the minimum saliency gain required for each token to be replaced. Then, by applying the Hungarian algorithm, we can find the optimal batch permutation $\sigma^*$ such that

$$\sigma^* = \arg\max_{\sigma} \sum_{i=1}^{n} C_{i,\sigma(i)}. \tag{7}$$

Then, by mixing $\tilde{X}^{(i)}$ and $X^{(\sigma^*(i))}$, our objective is optimized.

By incorporating sample difficulty assessment (section 3.1) into (5), our final objective is written as

$$\max_{\sigma \in \mathcal{M}} \max_{r_i \in \mathbb{B}^n} \sum_{i=1}^{b} \mathbb{1}_{\{\mathcal{F}(X^{(i)}, Y^{(i)}) < \tau\}} (P_{i,\sigma(i)} - \rho)^{\top} r_i$$
$$\text{s.t. } \mathbf{1}^{\top} r_i \leq n \times \mathbb{1}_{\{\mathcal{F}(X^{(i)}, Y^{(i)}) < \tau\}}, \tag{8}$$

where $\mathcal{F}$ is the difficulty measurer, and $\tau$ is the difficulty threshold. $\mathbb{1}$ denotes the indicator function, while $\mathbf{1}$ is a one vector. If the sample is hard, *i.e.*, $\mathcal{F}(X^{(i)}, Y^{(i)}) \geq \tau$, the constraint becomes $\mathbf{1}^{\top} r_i \leq 0$, enforcing all elements of $r_i$ to be 0. In that case, none of the tokens will be mixed. On the other hand, if the sample is easy, *i.e.*, $\mathcal{F}(X^{(i)}, Y^{(i)}) < \tau$, the constraint is $\mathbf{1}^{\top} r_i \leq n$ which is satisfied by any random $r_i$ vector. Then, the objective becomes identical to (5).

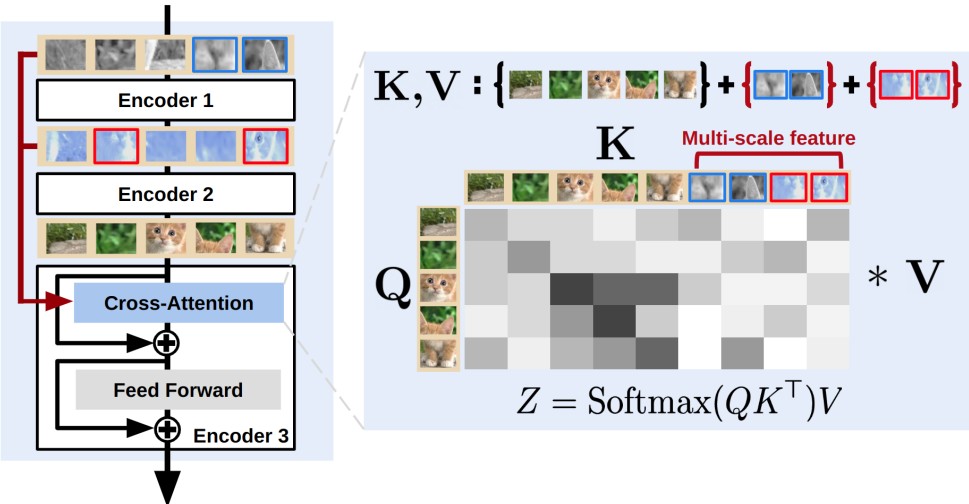

Figure 2: **Vertical TokenMixup**. By applying VTM, the $\kappa$ most salient tokens from each previous layer are brought up to form the key and value tokens. By utilizing a cross-attention mechanism, the dimension is preserved while being able to aggregate multi-scale information for token augmentation. The figure demonstrates the case when VTM is applied to layer 3.

### 3.3 Token-level Mixup

Now, the remaining question is *how* the tokens of $\tilde{X}^{(i)}$ and $X^{(\sigma^*(i))}$ are going to be mixed. As defined in objective (5), we use a hard replacement of tokens with respect to decision vector $r_i$. Then, mixing is performed by replacing token index $t$ from the original instance $i$, with token $t$ from the paired instance $j = \sigma^*(i)$, so that $P_{i,j}^{(t)} > \rho$ is satisfied. In other words, only the token indices that increase saliency level greater than $\rho$ are mixed. This is accomplished by defining an appropriate binary mask $M_t \in \mathbb{B}^n$ ($t = 1, 2, \ldots, n$) containing

$$M_t = \begin{cases} 0 & , \ S_t^{(j)} - \tilde{S}_t^{(i)} > \rho \\ 1 & , \ \text{otherwise} \end{cases} \tag{9}$$

where $\tilde{S}_t^{(i)}$ is the saliency score of $\tilde{X}_t^{(i)}$. Then, the mask is employed as

$$\tilde{X}_{\text{mix}}^{(i)} = M \odot \tilde{X}^{(i)} + (1 - M) \odot X^{(j)}, \tag{10}$$

where operator $\odot$ denotes element-wise multiplication and $\tilde{X}^{(i)}, X^{(j)}$ refer to the paired instances. We also reassign labels based on the replaced tokens' saliency score. The new mixed label is computed as

$$\tilde{Y}_{\text{mix}}^{(i)} = \frac{\sum_{t=1}^n M_t \cdot \tilde{S}_t^{(i)}}{\sum_{t=1}^n M_t \cdot \tilde{S}_t^{(i)} + (1 - M_t) \cdot S_t^{(j)}} \cdot \tilde{Y}^{(i)} + \frac{\sum_{t=1}^n (1 - M_t) \cdot S_t^{(j)}}{\sum_{t=1}^n M_t \cdot \tilde{S}_t^{(i)} + (1 - M_t) \cdot S_t^{(j)}} \cdot Y^{(j)}. \tag{11}$$

Consequently, relatively unimportant tokens will be replaced with salient tokens from another instance, and its label will be adjusted with respect to the change in overall saliency level. See Figure 1 for visualization, and Algorithm 1 for overall computation process. More detailed pseudocode is provided in the supplement.

### 3.4 Vertical TokenMixup

Mixing multiple *instances* is not the only way to mixup. We also present a simple variant of TokenMixup, which provides token-level augmentation within a single sample by utilizing the tokens from previous layers. Similar to (5), we can define a new objective for Vertical TokenMixup as

$$\max_{l \in \mathcal{L}} \ \max_{r_i \in \mathbb{B}^n} \ \sum_{i=1}^b (P_{i,l(i)} - \rho)^\top \cdot r_i, \tag{12}$$

where $\mathcal{L}$ is the set of previous layer indices, and $l(i)$ returns an arbitrary layer index for the $i^{th}$ instance. This objective can be optimized by a similar scheme as in Algorithm 1. However, considering that the tokens from each layer with an identical index tend to contain similar information, saliency difference matrix $P_{i,l(i)} = S_{l(i)} - S_i$ (as in (4)) will most likely be constant across indices. This may lead to meaningless mixup without much increase in saliency level. Therefore, we take a different approach to mix tokens *vertically*.

The simplest method would be to concatenate all tokens from previous layers and apply self-attention. But due to the quadratic complexity of self-attention, naive concatenation may induce excessive computation. To reduce overhead, we selectively pool the $\kappa$ most salient tokens from each previous layer and concatenate them to the original token set. Also, to preserve input dimension, we adopt the cross-attention mechanism. If $X_1$ is the original token set and $X_2$ is the concatenated token set, vertical cross-attention is expressed as

$$Z = \text{Softmax}(X_1 W_q (X_2 W_k)^\top) X_2 W_v, \tag{13}$$

where $W_q, W_k, W_v$ denote the projection layers for query, key, and value, respectively. See Figure 2 for visualization, and the supplement for pseudocode.

### 3.5 Discussions

**TokenMixup as Curriculum Learning.**   A general framework for curriculum learning consists of two main components: Difficulty Measurer and Training Scheduler [32]. TokenMixup satisfies these conditions, as ScoreNet measures the difficulty of each input, and augmenting the tokens based on input difficulty naturally schedules training. In the early training phase where the model parameter is not optimized, most instances will be evaluated *difficult* and no TokenMixup is performed. As the model converges, on the other hand, many samples will become *easy*, triggering augmentation to make the task more challenging overall. Also, TokenMixup is applied discriminatively based on individual sample difficulty which enables more intricate curriculum scheduling. See section 5.3 for empirical demonstration. These properties will be especially useful when other mixup methods [1–4] are used in parallel. For instance, if input mixup is already applied, further augmentation may be unnecessary. In such a case, ScoreNet will regard the sample sufficiently difficult, and no further mixup will be performed.

## 4   Experiments

### 4.1   Preliminaries

**Baseline.**   In experiments on CIFAR [33], we used Compact Convolution Transformer (CCT) [22] as our baseline. For ImageNet-1k [34] experiments, we used the vanilla ViT-B/16 [15] as baseline. Then, our augmentation methods are evaluated on three representative image classification datasets: CIFAR-10, CIFAR-100, and ImageNet-1K.

**Experimental setup.**   We evaluate TokenMixup and its variant, Vertical TokenMixup (VTM), on CIFAR and ImageNet with different settings. To avoid confusion, we denote the original TokenMixup as Horizontal TokenMixup (HTM) in the following sections. For CIFAR experiments, we adopt the 1500-epoch version of CCT. We modified the learning rate scheduler and positional embedding type to achieve better performance than original papers (denoted $*$ in Table 1). Other experiment settings follow [22], and all experiments on CIFAR datasets were conducted on a single RTX A6000 GPU. For ImageNet-1K experiments, we used the ViT-B/16 model pre-trained and fine-tuned on ImageNet-21k and ImageNet-1k, where we took advantage of the officially released pre-trained weights. Experiments for Horizontal TokenMixup were conducted on a single NVIDIA A100 GPU, and 4 RTX 3090 GPUs were used in parallel for Vertical TokenMixup. Other experiment settings are reported in the supplement.

### 4.2   Experimental Results

First, we demonstrate the image classification performance on CIFAR-10 and CIFAR-100 when TokenMixup is applied to CCT. From Table 1, we can see that applying Horizontal TokenMixup and Vertical TokenMixup simultaneously improves its strict CIFAR-100 baseline, by a significant margin of 0.70%. HTM and VTM each surpasses the baseline performance by a margin of 0.69% and 0.67% as well. Also, we outperform the previous state-of-the-art model (among from-scratch models) by a

Table 1: **Experimental results on CIFAR.** Experiment results with TokenMixup on CCT are compared with state-of-the-art models that do not use pretrained models for initialization. Top-1 validation accuracy is reported for each model, and baseline models denoted with * reports retrained performance. Also, the number in the parenthesis indicates the number of epochs used for training, and the parameter numbers are retrieved from the CIFAR-100 models. Best performances are highlighted in yellow.

| Models | # Params | MACs | CIFAR-10 | CIFAR-100 |
|---|---|---|---|---|
| *Convolutional Network based* | | | | |
| ResNet18 | 11.18 M | 0.04 G | 90.27 | 63.41 |
| ResNet50 | 23.53 M | 0.08 G | 90.60 | 61.68 |
| ResNet1001-v2 [35] | 10.33 M | 1.55 G | 95.08 | 77.29 |
| MobileNetV2/2.0 [36] | 8.72 M | 0.02 G | 91.02 | 67.44 |
| WRN-28-10 [37] | 36.5 M | - | 96.00 | 80.75 |
| WRN-40-4 [37] | 8.9 M | - | 95.47 | 78.82 |
| ResNeXt-29-8×64d [38] | 34.4 M | - | 96.35 | 82.23 |
| ResNeXt-29-16×64d [38] | 68.1 M | - | 96.42 | 82.69 |
| *Transformer based* | | | | |
| ViT-Lite-6/4 [22] | 3.19 M | 0.22 G | 93.98 | 73.33 |
| ViT-Lite-7/4 [22] | 3.72 M | 0.26 G | 93.57 | 73.94 |
| NesT-T [39] | 17.0 M | - | 96.04 | 78.69 |
| NesT-B [39] | 68.0 M | - | 97.20 | 82.56 |
| CVT-6/4 [22] | 3.19 M | 0.22 G | 93.60 | 74.23 |
| CCT-7/3x1(1500) [22] | 3.76 M | 0.95 G | 97.48 | 82.72 |
| CCT-7/3x1(1500) * | 3.78 M | 0.95 G | 97.48 | 82.87 |
| CCT-7/3x1(1500) + **Horizontal TM (ours)** | 3.81 M | 0.95 G | **97.57** | **83.56** |
| CCT-7/3x1(1500) + **Vertical TM (ours)** | 3.78 M | 0.95 G | 97.78 | 83.54 |
| CCT-7/3x1(1500) + **HTM + VTM (ours)** | 3.81 M | 0.95 G | **97.75** | 83.57 |

Table 2: **Experimental results on ImageNet.** Experiment results with TokenMixup on ViT are compared. We report Top-1 and Top-5 validation accuracy on ImageNet-1k. For the baseline model, we used ViT-B/16 $(224 \times 224)$. $^\dagger$ denotes fine-tuned performance officially reported by the authors of ViT.

| Models | # Params | MACs | Top-1 Acc. | Top-5 Acc. |
|---|---|---|---|---|
| *Convolutional Network based* | | | | |
| ResNet50 [40] | 25.5 M | 4.3 G | 76.20 | - |
| ResNet152 [40] | 60.19 M | 11.58 G | 78.57 | - |
| DenseNet-264 [41] | - | - | 79.20 | 94.71 |
| WRN-50-2-bottleneck [37] | 68.9 M | - | 78.10 | 93.97 |
| ReGNetY-4G [42] | 21 M | 2.0 G | 80.0 | - |
| ResNeXt-50-2×40d [38] | - | - | 77.00 | - |
| ResNeXt-101-64x4d [38] | - | - | 79.60 | 94.70 |
| *Transformer based* | | | | |
| ViT-B/16-224$^\dagger$ [15] | 86.6 M | 16.9 G | 81.2 | - |
| ViT-B/16-224 + **Horizontal TM (ours)** | 87.3 M | 16.9 G | 82.37 | **96.29** |
| ViT-B/16-224 + **Vertical TM (ours)** | 86.6 M | 16.9 G | **82.30** | 96.33 |
| ViT-B/16-224 + **HTM + VTM (ours)** | 87.3 M | 16.9 G | **82.32** | **96.27** |

huge margin of 0.85%. In the case of CIFAR-10, we achieved higher accuracy of 97.57% and 97.78% with HTM and VTM, respectively. Considering the highly saturated performance on this dataset, a 0.30% improvement over the baseline is not trivial.

Performance on ImageNet-1K is evaluated on the vanilla ViT [15] model. The officially reported Top-1 accuracy is 81.2%, which is achieved by using weights pre-trained on ImageNet-21K and fine-tuning on ImageNet-1K. By fine-tuning the model further by employing our augmentation methods, we achieve Top-1 accuracy of 82.37% with Horizontal TokenMixup, 82.30% with Vertical TokenMixup, and 82.32% by using both. Also note that HTM marginally increases parameter size by only 0.7M, while VTM does not require any additional parameters. See the supplement for further experiments on other baselines.

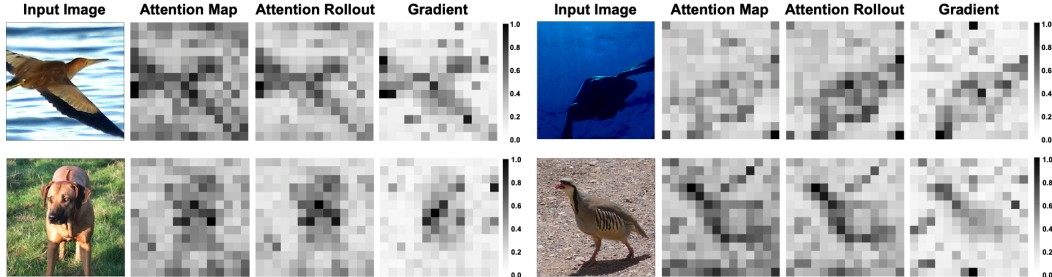

Figure 3: **Qualitative comparison of saliency detectors.** The saliency map derived for each method is provided. We can observe that the methods render similar outputs. Note, the values are minmax scaled to [0,1].

Table 3: **Saliency detector comparison.** Our method (HTM) achieves high accuracy while being ×15 as fast as the gradient method. Also, we observe only ×3 latency compared to the random baseline.

| Saliency type | **Gradient-based** | **Attention-based** | **Random baseline** |
|---|---|---|---|
| Avg. Latency (ms) | 236 (×14.8) | 16 (×1.0) | 5 (×0.3) |
| CIFAR-100 Acc. | 83.28 | **83.56** | 83.05 |

# 5 Analysis

Here, we analyze TokenMixup to answer the following research questions: **Q1**. Is the Attention an appropriate saliency approximation? **Q2**. What are the effects of each component in TokenMixup? **Q3**. What properties does TokenMixup have?

## 5.1 Soundness of Saliency Estimation

In TokenMixup, we estimate the saliency using the attention map from the subsequent layer. The soundness of this approximation would be one of the main questions regarding saliency detection. So, we provide relevant analyses on the appropriateness of our saliency approximation method to answer the research question **Q1**.

**Qualitative examples.** Here, we provide a qualitative analysis on the performance of each saliency detector. In Figure 3, we can observe that there is no significant difference between the full attention rollout and the 1-step approximation, *i.e.*, the subsequent attention map. Slight discrepancy in the sharpness of the distribution can be observed, but it does not greatly affect our method that utilizes saliency threshold $\rho$. Moreover, attention-based methods seem to be more accurate than the gradient-based detector in certain cases. See the supplement for additional qualitative examples, and also the quantified comparison on the sharpness measures of the two methods.

**Efficiency comparison.** We compare the efficiency of three saliency detectors: gradient-based, attention-based, and random baseline. For the gradient-based detector, we follow the method used in Co-Mixup [4], and random baseline refers to a random selection of tokens. In Table 3, we report the average latency of saliency detection per iteration in our CIFAR-100 experiment setting with a batch size of 128. Compared to the gradient-based method, attention-based saliency detection with 1-step rollout achieves approximately ×15 speed-up while demonstrating even better performance.

## 5.2 Component Analysis

In this section, we analyze the effect of each component in Horizontal TokenMixup and Vertical TokenMixup so as to answer **Q2**. In Table 4, we provide ablation studies on components of HTM: Sample difficulty assessment (ScoreNet), Optimal assignment, Token-level mixup, and Saliency-based label reassignment. By ablating ScoreNet, all instances are mixed regardless of sample difficulty; note, ScoreNet still receives supervision signals. By getting rid of optimal assignment, mixup pairs are not assigned via Hungarian matching; pairs are randomly selected. Token-level mixup and label reassignment is relevant to the method by which tokens are selected. Then, by ablating mixup, we randomly select tokens to replace, while label reassignment ablation derives new labels based on the number of tokens that have been replaced. We also conducted sensitivity tests on key hyperparameters $\tau$, $\rho$, and $\kappa$, but tables and discussions were moved to the supplement due to spatial constraints.

Table 4: **Ablation study on CIFAR-100.** Performance on CIFAR-100 is evaluated by ablating each component one-by-one. We observe that optimal assignment and label reassignment plays a significant role.

| Ablated Component | CIFAR-100 Accuracy |
|---|---|
| CCT-7/3x1 + HTM | **83.56** (- 0.00) |
| (–) Sample difficulty assessment (ScoreNet) | 82.85 (- 0.71) |
| (–) Optimal assignment | 82.61 (- 0.95) |
| (–) Token-level mixup | 83.05 (- 0.51) |
| (–) Saliency-based label reassignment | 82.62 (- 0.94) |

Table 5: **Mixup combinations.** CCT performance on CIFAR-100 is compared for mixup method combinations. * denotes values reported in the original paper. Best performances among comparable settings are highlighted.

| Mixup Combination | CIFAR-100 Accuracy |
|---|---|
| CCT-7/3x1 | 76.80 |
| CCT-7/3x1 + Manifold Mixup | 76.72 |
| CCT-7/3x1 + **Horizontal TM** | 77.34 |
| CCT-7/3x1 + **Vertical TM** | 78.18 |
| CCT-7/3x1 + Input Mixup & Cutmix * | 82.87 |
| CCT-7/3x1 + Input Mixup & Cutmix + Manifold Mixup | 80.08 |
| CCT-7/3x1 + Input Mixup & Cutmix + **Horizontal TM** | **83.56** |
| CCT-7/3x1 + Input Mixup & Cutmix + **Vertical TM** | **83.54** |
| CCT-7/3x1 + Input Mixup & Cutmix + **HTM + VTM** | **83.57** |

## 5.3   What TokenMixup Learns

Here, we answer our research question **Q3** by analyzing what our mixup methods are learning. We demonstrate a curriculum learning perspective in terms of HTM, and compare our method with a direct mixup alternative, manifold mixup [6].

**A Curriculum Learning perspective.**   In Figure 4 we plot the trend in the number of samples per batch TokenMixup is activated based on ScoreNet. In the first few epochs (regime (A)), the number of mixed samples is relatively low, as there is less need for augmentation in the early phase of learning. But as the learnable parameters warm up, the number increases abruptly. In regime (B), the average number of mixed tokens slowly increases as model performance steadily improves. In the last learning phase (regime (C)), mixup frequency converges.

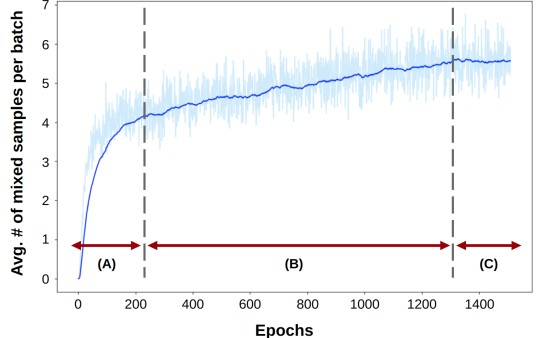

Figure 4: **Trend in the number of mixed instances.**

**Comparison with Manifold Mixup.**   TokenMixup is applied to the intermediate tokens, similarly to manifold mixup [6] which is an input mixup [1] method applied to intermediate features. By using manifold mixup, transformer tokens are mixed via linear interpolation, instead of the hard replacement of tokens as in TokenMixup. In Table 5, our methods are compared with manifold mixup, along with various combinations of original mixup methods (*i.e.*, input mixup and cutmix). The best setting in the CCT paper adopted input mixup and cutmix. By applying HTM and VTM on top of that setting, we achieved state-of-the-art performance. On the other hand, by applying manifold mixup, we observed severe deterioration in accuracy. We conjecture that soft mixing of transformer tokens leads to over-smoothing of features, which may be inadequate for self-attention layers. See the supplement for visual aid.

## 6   Conclusion

We proposed TokenMixup for transformers, which adopts the attention map in place of computationally heavy gradient-based saliency detectors. By adopting the attention as the saliency map, we achieve ×15 speed-up compared to the gradient-based method with even better performance. We also introduce a novel perspective of curriculum learning to mixup, which enables adaptive augmentation based on sample difficulty and training schedule. To add diversity, Vertical TokenMixup is introduced, which mixes tokens from different layers for multi-scale feature augmentation within a single sample.

With TokenMixup, we achieve state-of-the-art performance on CIFAR-100, and improve baselines by significant margins on CIFAR-10 and ImageNet-1K.

## Acknowledgments and Disclosure of Funding

This work was partly supported by ICT Creative Consilience program (IITP-2022-2020-0-01819) supervised by the IITP; the National Supercomputing Center with supercomputing resources including technical support (KSC-2022-CRE-0100) and Kakao Brain Corporation.

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
