# TokenMixup: Efficient Attention-guided Token-level Data Augmentation for Transformers (Supplementary Material)

**Overview.** We provide additional materials that were omitted from the main paper due to limited space. In section A, additional figures and algorithms are presented. In section B, we provide hyperparameter settings and further experimental details, and sensitivity tests on key hyperparameters $\tau, \rho$, and $\kappa$ are conducted in section C. In section D, additional qualitative examples for Figure 3 in the main paper are listed. In section E, a quantitative comparison of saliency maps is given. In section F, we further provide experiment results on different vision transformer architectures. In section G, we compare our methods with two different random baseline models. In section H, analysis on ScoreNet is presented. In section I, robustness of TokenMixup is compared with its baseline model. In section J, limitations & negative societal impacts are discussed. In section K, qualitative analysis results are provided for Manifold Mixup.

## A  Additional Figures and Algorithms

### A.1  Additional Figures

Two types of ScoreNet (main paper section 3.1) architecture are presented in Figure A.1. The left architecture is used for CCT [1] (CIFAR experiments), and the right is used for ViT [2] (ImageNet experiments).

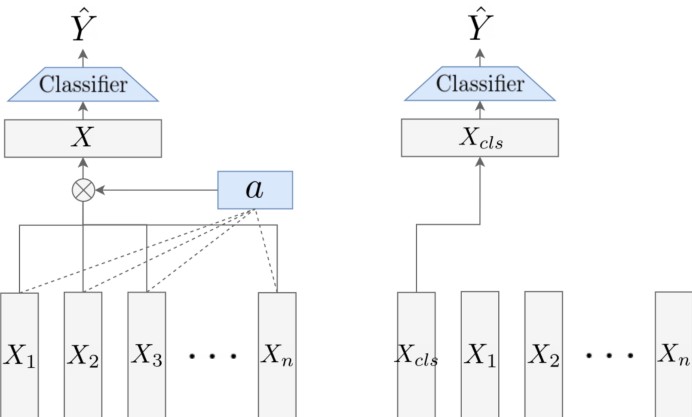

Figure 1: **ScoreNet Architecture.** (left) ScoreNet for transformers that do not use *cls* tokens. Each token is projected to a scalar value which serves as the attention weight for linear combination. Then, the linearly-combined token $X$ is used for classification. (right) If *cls* token exists, we directly use the token for classification.

## A.2 Additional Algorithms

---

**Algorithm 1** TokenMixup – Full Version

---

**Input:** $X \in \mathbb{R}^{b \times n \times d}$, $Y \in \mathbb{R}^{b \times c}$, $\tau$, $\rho$
**Output:** $Z \in \mathbb{R}^{b \times n \times d}$, $Y$, $\hat{Y} \in \mathbb{R}^{b \times c}$

  1: $\hat{Y} \leftarrow \text{ScoreNet}(X)$
  2: $U \leftarrow \text{CrossEntropy}(\text{Softmax}(\hat{Y}), Y)$         ▷ Eq. (1)
  3: $A \leftarrow$ apply $\ell$-step attention rollout         ▷ Eq. (2)
  4: $S_t \leftarrow \frac{1}{n} \sum_{i=1}^{n} A_{i,t}$    s.t. $t = 1, 2, \ldots, n$         ▷ Eq. (3)
  5: $\text{idx} \leftarrow$ sample indices where $U < \tau$
  6: $b' \leftarrow$ number of items in idx
  7: $\tilde{X}, \tilde{Y}, \tilde{S} \leftarrow X[\text{idx}], Y[\text{idx}], S[\text{idx}]$
  8: $C_{ij} \leftarrow \sum_t \max(S_t^{(j)} - \tilde{S}_t^{(i)} - \rho,\ 0)$    s.t. $i = 1, 2, \ldots, b'$ and $j = 1, 2, \ldots, b$     ▷ Eq. (6)
  9: $\sigma(m) \leftarrow \text{HungarianMatching}(C_{ij})$    s.t. $m = 1, 2, \ldots, b'$ and $\sigma(m) \in \{1, 2, \ldots, b\}$   ▷ Eq. (7)
10: **for** sample index $m = 1, 2, \ldots, b'$ **do**
11:     $M_t \leftarrow 0$  if $S_t^{(\sigma(m))} - \tilde{S}_t^{(m)} > \rho$ else 1    s.t. $t = 1, 2, \ldots, n$     ▷ Eq. (9)
12:     $\tilde{X}^{(m)} \leftarrow M \odot \tilde{X}^{(m)} + (1 - M) \odot X^{(\sigma(m))}$    s.t. $M = [M_t]_{t=1}^{n}$     ▷ Eq. (10)
13:     $\tilde{Y}^{(m)} \leftarrow \frac{\sum_t M_t \cdot \tilde{S}_t^{(m)}}{\sum_t M_t \cdot \tilde{S}_t^{(m)} + (1 - M_t) \cdot S_t^{(\sigma(m))}} \tilde{Y}^{(m)} + \frac{\sum_t (1 - M_t) \cdot S_t^{(\sigma(m))}}{\sum_t M_t \cdot \tilde{S}_t^{(m)} + (1 - M_t) \cdot S_t^{(\sigma(m))}} Y^{(\sigma(m))}$     ▷ Eq. (11)
14: **end for**
15: $X \leftarrow X[\neg \text{idx}] \cup \tilde{X}$    s.t. $\tilde{X} = \{\tilde{X}^{(m)}\}_{m=1}^{b'}$
16: $Y \leftarrow Y[\neg \text{idx}] \cup \tilde{Y}$    s.t. $\tilde{Y} = \{\tilde{Y}^{(m)}\}_{m=1}^{b'}$
17: $Z \leftarrow \text{MultiHeadSelfAttention}(X)$
18: **return** $Z, Y, \hat{Y}$

---

**Algorithm 2** Vertical TokenMixup

---

**Input:** $X \in \mathbb{R}^{b \times n \times d}$, $Y \in \mathbb{R}^{b \times c}$, $\kappa$
**Output:** $Z \in \mathbb{R}^{b \times n \times d}$

  1: $L \leftarrow$ previous layer indices
  2: **for** layer index $l \in L$ **do**
  3:     $X^{(l)} \leftarrow$ token input for layer $l$
  4:     $A^{(l)} \leftarrow$ apply $\ell$-step attention rollout     ▷ Eq. (2)
  5:     $S_t \leftarrow \frac{1}{n} \sum_{i=1}^{n} A_{t,i}^{(l)}$    s.t. $t = 1, 2, \ldots, n$     ▷ Eq. (3)
  6:     $X^{(l)} \leftarrow$ select top $\kappa$ tokens w.r.t $S_t$
  7: **end for**
  8: $X' \leftarrow \bigcup_{l \in L'} X^{(l)}$    s.t. $L' = \{l_0\} \cup L$ and $l_0 = $ current layer index
  9: $Z \leftarrow \text{MultiHeadCrossAttention}(\text{query} = X, \text{key} = X', \text{value} = X')$     ▷ Eq. (13)
10: **return** $Z$

---

Algorithm 1 contains a more detailed pseudocode of (Horizontal) TokenMixup, which is provided in Algorithm 1 of the main paper. In line 1-2, the difficulty score $U \in \mathbb{R}^b$ is computed with ScoreNet, followed by the $\ell$-step attention rollout ($\ell = 0$ in our case) to retrieve the saliency score of each token from each sample instance in line 3-4. In line 5-7, *easy* instances are identified with respect to $U$ and $\tau$. Then, Hungarian matching is performed in line 8-9, resulting in an optimally assigned pair with respect to $C$ in line 8. Actual mixup is performed in line 10-14 and the mixed instances are concatenated with the un-mixed instances in line 15-16, which corresponds to Algorithm 1 - line 8 in the main paper.

Algorithm 2 presents the pseudocode for Vertical TokenMixup. In line 1, previous layer indices are retrieved, *e.g.*, $L = [1, 2, 3, 4]$ if VTM is applied to the $5^{th}$ layer. Then, saliency score $S_t$ is computed per layer, similarly to Algorithm 1 line 3-5. In line 6, $X^{(l)} \in \mathbb{R}^{b \times \kappa \times d}$ are selected, which are concatenated with the input tokens $X^{(l_0)} = X$ to form $X' \in \mathbb{R}^{b \times (n + \kappa |L|) \times d}$, which will be projected to key and value tokens. Finally, in line 9, cross-attention is performed.

# B  Hyperparameters

## B.1  Compact Convolutional Transformer

The hyperparameter settings largely follow the conditions in the original paper of CCT [1]. In the case of CIFAR-10 experiments, we use the identical settings used in the original paper except for the positional embedding. While the baseline model uses the sinusoidal positional embedding, we instead use the learnable positional embedding. This is also the case for CIFAR-100, which we further tune the learning rate scheduler. We set a different learning rate scheduler starting from 6e-4 down to 1e-5 with consine annealing. We identically train the model for 1500 epochs with an additional 10-epoch cool-down period. Finally, we applied HTM to the $3^{rd}$ layer for both CIFAR-10 and CIFAR-100, and applied VTM to the $2^{nd}$ and $3^{rd}$ layer respectively. When applying both, HTM and VTM is applied to the $2^{nd}$ and $3^{rd}$ layer for CIFAR-10, and the $3^{rd}$ and $2^{nd}$ layer for CIFAR-100.

## B.2  Vision Transformer

As mentioned in the main paper, we experimented Horizontal TokenMixup and Vertical TokenMixup on different GPU environments for ImageNet-1K. In the case of HTM, ViT [2] was trained on a single NVIDIA A100 GPU with a batch size of 504. The learning rate was scheduled with cosine annealing with a maximum learning rate 0.015 to a minimum value of 0.0015. For VTM, we used 4 RTX 3090 GPU's in parallel also with total batch size of 504 ($126 \times 4$). The learning rate is scheduled from 3e-2 to 3e-3 with cosine annealing. In the case of ImageNet, HTM and VTM was both applied to the $4^{th}$ layer, which was determined empirically. When used together, HTM was applied to the $4^{th}$ and VTM to the $5^{th}$. For all cases, the model was fine-tuned for 30 epochs, and the SGD optimizer is adopted. In terms of architecture, we found that applying stop-grad on the input of ScoreNet renders more stable results when fine-tuning.

# C  Hyperparameter Sensitivity Tests

$\tau$ (main paper section 3.1) refers to the sample difficulty threshold, where setting low $\tau$ leads to overestimation of sample difficulty, and *vice versa*. From left Table 1 we found $\tau = 0.2$ to be optimal, and by setting $\tau = 0$, all samples were regarded difficult and none of the instances had TokenMixup applied. In the other extreme with infinite $\tau$, *i.e.* TokenMixup applied to all instances, performance of 82.85 was recorded. This represents the case where the ScoreNet takes no effect.

$\rho$ (main paper section 3.2), on the other hand, refers to the saliency difference threshold. That is, $\rho$ controls the minimum amount of saliency gain required for a token to be replaced. By setting $\rho = 0$, tokens are mixed in a way that maximizes total saliency. If $\rho$ is maximal, no tokens are mixed, as shown in middle Table 1.

Finally, $\kappa$ (main paper section 3.4) is the number of tokens to be pooled from each previous layers when VTM is adopted. We did not observe specific trends or patterns by controlling $\kappa$. We could see that model performance is quite robust to this hyperparameter.

Table 1: **Sample difficulty and saliency thresholds.** (left) CIFAR-100 performance with respect to sample difficulty threshold $\tau$ is reported. We also report the average number of samples HTM is applied. (middle) HTM performance with respect to saliency difference threshold $\rho$ is reported. We also report the average number of tokens mixed after convergence. (right) VTM performance by token sampling number per layer, $\kappa$, is reported.

| $\tau$ | Accuracy | Avg. Count | $\rho$ | Accuracy | Avg. Count | $\kappa$ | Accuracy |
|--------|----------|------------|--------|----------|------------|----------|----------|
| 0.00 | 83.13 | 0 | 0.000 | 83.13 | 116.01 | 0 | 82.87 |
| 0.10 | 83.30 | 29 | 0.001 | 83.51 | 78.76 | 5 | 83.16 |
| 0.15 | 83.07 | 616 | 0.003 | 83.17 | 31.87 | 10 | 83.01 |
| 0.20 | **83.56** | 2,142 | 0.005 | **83.56** | 14.89 | 30 | 83.09 |
| 0.25 | 83.30 | 5,471 | 0.007 | 83.19 | 7.64 | 50 | 83.28 |
| 0.30 | 82.91 | 11,072 | 0.010 | 82.75 | 1.46 | 80 | **83.54** |
| max | 82.85 | 50,000 | max | 82.90 | 0.0 | 100 | 83.46 |

# D  Additional Qualitative Results

Here, we provide an extended list of the qualitative comparison of the saliency detector outputs, presented in section 5.1.

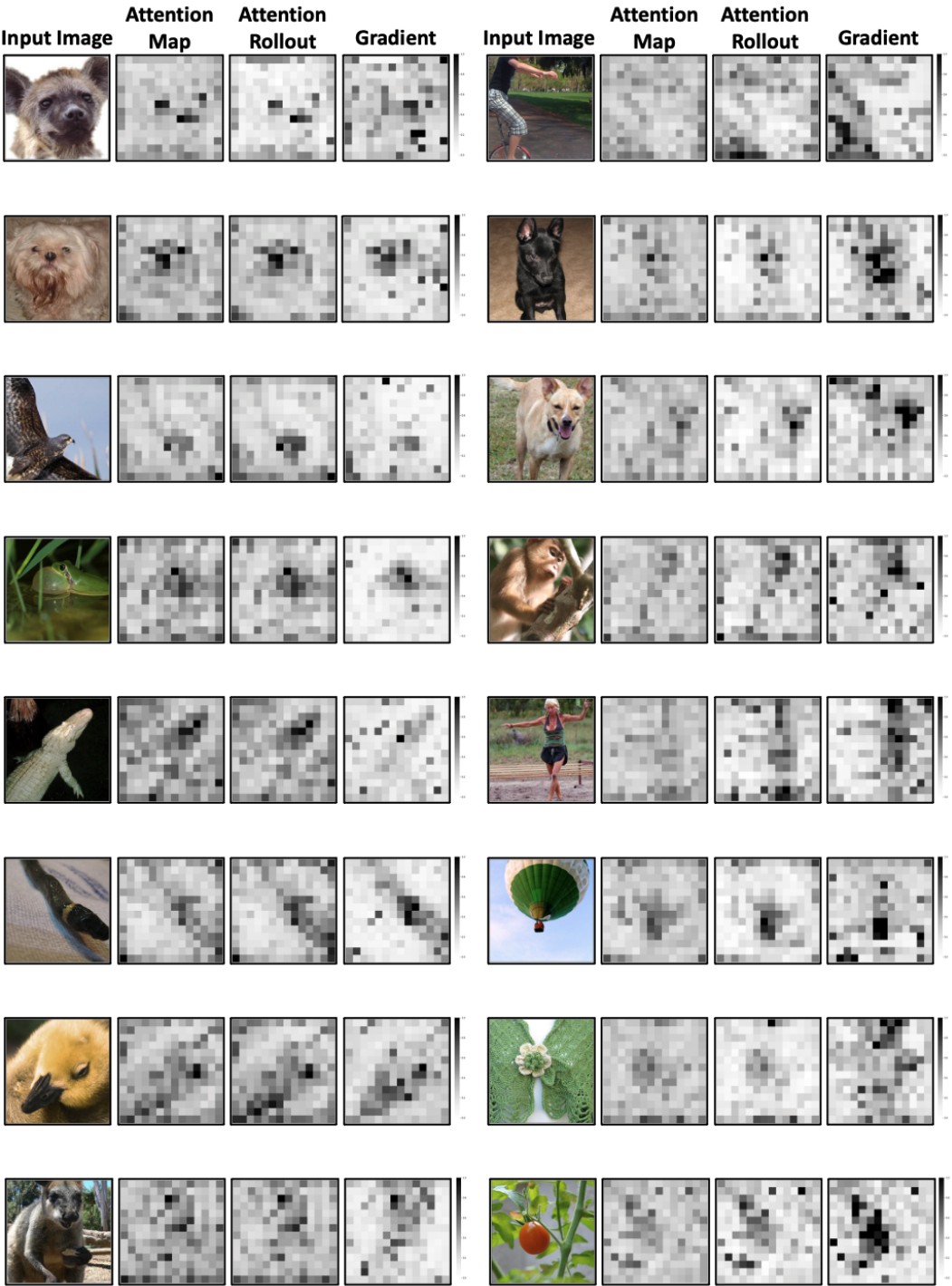

Figure 2: **Additional qualitative examples**

# E  Saliency Map Sharpness Comparison

In our qualitative analysis, we observed that gradient-based saliency maps occasionally have irregular representation of the input image, while attention-based saliency maps are usually smoother than gradient maps. To quantitatively demonstrate this tendency, we here compare the smoothness statistics of the two saliency maps. In Table 2, we provide the average Total Variation and Variance of the saliency map. Total variation is derived either with the L1 or L2 norm, each computed as

$$\text{TV}_{L1} = \sum_{i,j} |S_{i+1,j} - S_{i,j}| + |S_{i,j+1} - S_{i,j}|$$

$$\text{TV}_{L2} = \sum_{i,j} \sqrt{|S_{i+1,j} - S_{i,j}|^2 + |S_{i,j+1} - S_{i,j}|^2}$$

where $S$ is the attention or gradient-based saliency map. Also note that both saliency maps were normalized to sum to 1, before computing the measures.

Table 2: **Saliency map distribution sharpness comparison.**

|  | Attention-based | Gradient-based |
|---|---|---|
| Total Variation (L1) | 0.5454 | 1.3176 |
| Total Variation (L2) | 0.1441 | 0.3910 |
| Variance | 4.88E-6 | 1.95E-5 |

# F  Comparison with other Vision Transformer Architectures

Here, we provide additional experiments on other transformer architectures. By applying TokenMixup to ViT-Lite-7/4 [1] with CIFAR-100, we achieve the best performance by utilizing HTM and VTM simultaneously. On the other hand, we also experimented on PVTv2-B0 [3] for ImageNet-1k. By using HTM, we achieve higher accuracy. Note, PVTv2 projects the key/value tokens to a smaller token set, which requires a simple attention map rescaling trick to find the salient tokens for HTM. For VTM, however, this trick is not directly applicable due to different feature dimensions of layers.

Table 3: **Experiments on other architectures.** TokenMixup methods are applied to other vision transformer architectures including ViT-Lite and PVTv2. * denotes the reproduced result.

| model | CIFAR-100 Accuracy |
|---|---|
| ViT-Lite-7/4 (1500) * [1] | 79.44 |
| ViT-Lite-7/4 (1500) + HTM | 80.53 |
| ViT-Lite-7/4 (1500) + VTM | 80.44 |
| ViT-Lite-7/4 (1500) + HTM + VTM | **80.65** |

| model | ImageNet-1k Top1 | ImageNet-1k Top5 |
|---|---|---|
| PVTv2-B0 [3] | 70.46 | 90.16 |
| PVTv2-B0 + HTM | **71.20** | **90.43** |
| PVTv2-B0 + VTM | - | - |

# G  Random Baseline Comparisons

## G.1  Random Sample Selection

We compare (Horizontal) TokenMixup with a baseline which randomly selects the samples from a uniform distribution to apply HTM. We used the average sampling number of 5 for this experiment on CIFAR-100. Note, the latency measures the average latency for each iteration (not just the sampling module).

Table 4: **Comparison with random sample selection**

| Mixup type | Accuracy | Latency (ms) |
|------------|----------|--------------|
| Random | 82.12 | 85 |
| HTM (Ours) | **83.56** | 89 |

## G.2  Random Token Selection

We compare HTM with a baseline that matches pairs randomly for mixup, computes the average number of salient tokens with respect to $\rho = 0.005$, and randomly selects the corresponding amount of tokens from a uniform distribution to apply our method. For VTM, we use a baseline that also randomly selects tokens from a uniform distribution. In Table 5, Horizontal TokenMixup and Vertical TokenMixup both outperform the random baseline settings.

Table 5: **Comparison with random token selection**

| Mixup type | **Horizontal** | **Vertical** |
|------------|----------------|--------------|
| Random | 83.05 | 83.33 |
| Ours | **83.56** | **83.54** |

# H  On the Accuracy of ScoreNet Evaluation

In general, sample difficulty is assessed by the prediction error at the final layer. In this work, however, we measure it at an intermediate layer where we apply (Horizontal) TokenMixup. By doing so, we observed improvement in performance as demonstrated in Table 6. Also, there is a 36% speed-up in terms of latency, since we no longer need to take additional layer propagation steps to retrieve the difficulty score. Note, we measured the computation time per iteration assuming that mixup is applied to *all* sample instances.

Table 6: **ScoreNet position analysis**

| ScoreNet position | Accuracy | Latency* (ms) |
|-------------------|----------|---------------|
| final layer | 83.11 | 175 |
| Ours | **83.56** | 119 |

# I  Robustness of TokenMixup

Here, we present experiments on our methods' robustness to corrupted examples. We use the Gaussian noise $\epsilon_{i,j} \sim N(0, \sigma^2)$ as the corruption, which is applied with $X'_{i,j} = X_{i,j} + \epsilon_{i,j}$ where $X_{i,j}$ denotes the pixel values of image $X$ at position $(i, j)$. We demonstrate how the performance decays as corruption intensifies. We only report up to $\sigma = 0.5$ since standard deviaion greater than 0.5 was too strong, which we find meaningless to report. From Table 7, performance of each model with different settings is reported, and the values in parentheses indicate the performance decay compared to the original setting with $\sigma = 0$. We can observe from the value that Horizontal TokenMixup is generally robust across different intensities. For each intensity level, the setting with the least performance decay is shown in bold.

We further present robustness tests on adversarial examples. In Table 8, experiment results with PGD attack [4] are reported. Gradient step size $\alpha$ is controlled to observe how performance decays. For each step size, the setting with the least performance decay is shown in bold. We could observe that Horizontal TokenMixup again shows the best robustness overall, except for the case when $\alpha = 0.03$.

Table 7: **Robustness to Gaussian noise**

| $\sigma$ | 0.0 | 0.1 | 0.2 | 0.3 | 0.4 | 0.5 |
|---|---|---|---|---|---|---|
| baseline | 82.87 (-0.00) | 78.10 (-4.77) | 68.99 (-13.88) | 55.73 (-27.14) | 41.58 (-41.29) | 29.79 (-53.08) |
| baseline + **Horizontal TM** | **83.56** (-0.00) | **79.11** (-4.45) | **71.60** (-11.96) | **60.71** (-22.85) | **45.95** (-37.61) | **32.34** (-51.22) |
| baseline + **Vertical TM** | 83.54 (-0.00) | 78.68 (-4.86) | 70.21 (-13.33) | 58.24 (-25.30) | 43.94 (-39.60) | 32.09 (-51.45) |

Table 8: **Robustness to PGD attack**

| $\alpha$ | none | 0.001 | 0.003 | 0.010 | 0.030 |
|---|---|---|---|---|---|
| baseline | 82.87 (-0.00) | 69.68 (-13.19) | 64.54 (-18.33) | 48.72 (-34.15) | **30.05** (-52.82) |
| baseline + **Horizontal TM** | **83.56** (-0.00) | **70.94** (-12.62) | **66.05** (-17.51) | **50.70** (-32.86) | 30.04 (-53.52) |
| baseline + **Vertical TM** | 83.54 (-0.00) | 70.47 (-13.07) | 65.13 (-18.41) | 49.40 (-34.14) | 29.59 (-53.95) |

# J  Limitations and Potential Negative Societal Impacts

Transformers are flexible attention-based models that are widely adopted across domains. Considering the computational complexity of transformers due to the huge number of parameters, we proposed TokenMixup as an efficient token-level mixup method specifically designed for transformer-based models. Thus, it is difficult to extend our method to models, other than transformers, which do not utilize attention layers. In such cases, gradient-based saliency detectors need to be used. However, if any sort of attention value for each token/pixel can be derived with additional modules (*e.g.* SE-Net [5], Residual Attention Network [6], CBAM [7], Non-local neuralnet [8], Gcnet [9]), TokenMixup can be applied.

As TokenMixup is a simple token augmentation method for transformer-based models, we do not expect any negative societal impacts directly by this work.

# K   Analysis on the failure of Manifold Mixup

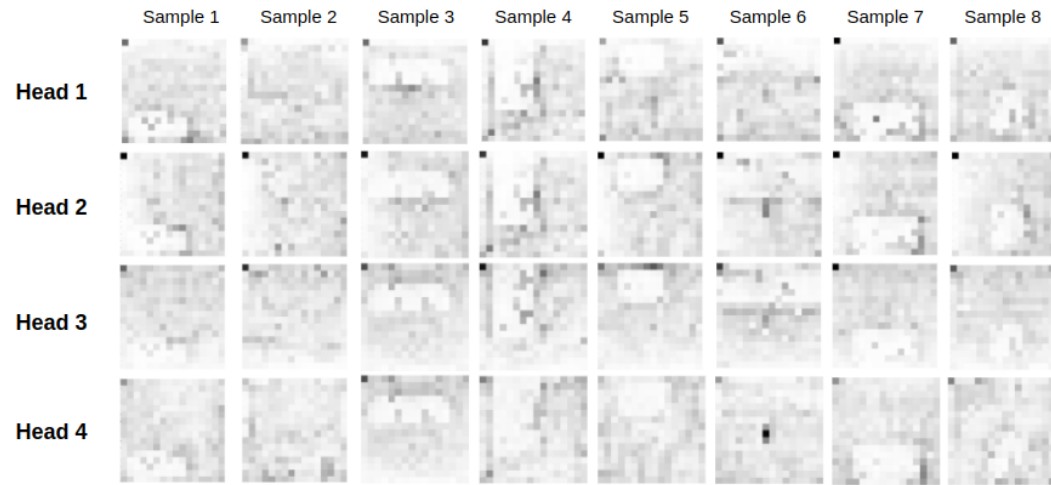

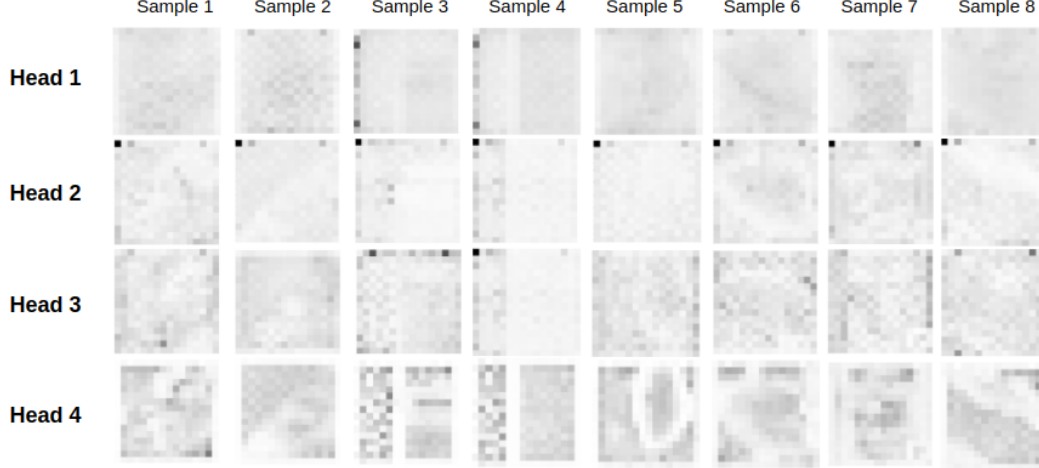

Figure 3: **Attention map comparison of HTM and Manifold Mixup.** 8 mixed pairs are sampled each for Horizontal TokenMixup and Manifold Mixup, to compare the attention map from the subsequent layer. Artifacts and smoothed attention heads (*e.g.* head 1 & 2) are observed for Manifold Mixup, which we believe is the reason Manifold Mixup did not perform well in our experiments.

# L   Miscellaneous

- Source code for Compact Convolutional Transformer [1] is released under Apache License[1].
- Source code we used for Vision Transformer [2] is released under MIT License[2], and the pretrained weights are officially published under Apache License[3].
- We used official benchmark datasets for evaluation, which does not contain offensive content.

---

[1]https://github.com/SHI-Labs/Compact-Transformers
[2]https://github.com/jeonsworld/ViT-pytorch
[3]https://github.com/google-research/vision_transformer