# OpenReview forum: "TokenMixup: Efficient Attention-guided Token-level Data Augmentation for Transformers"
_NeurIPS.cc/2022/Conference — NeurIPS 2022 Accept_

### Official Review · Reviewer_pM5D · 2022-07-03

**Rating:** 6
**Confidence:** 5
**Soundness:** 3 good
**Presentation:** 3 good
**Contribution:** 3 good

**Summary:**

This paper presents an efficient attention-guided data augmentation method, named TokenMixup. As the name implies, the goal is to provide an efficient mixup method based on attentions to improve the performance of classification models. Vertical TokenMixup is also introduced, which considers using tokens from different feature levels for mixup. Experiments on CIFAR and ImageNet show that the proposed approach performs better than the original mixup method.

**Questions:**

More analysis on the ImageNet dataset would make the analysis more convincing.

For more questions, please refer to the Strengths And Weaknesses part.

**Limitations:**

Limitations have been mentioned in the supplementary material. No potential negative societal impact is found.

**Strengths And Weaknesses:**

Overall speaking, the writing quality of this paper is good and it is easy to follow. Some strengths and weaknesses of this paper are listed as follows.

Pros:

- The idea of this paper is interesting. Unlike previous follow-ups of the mixup method, this paper follows the curriculum learning framework and introduces a tiny scorenet to measure the saliency of this token. Besides, the Vertical TokenMixup method is also interesting. This is the first work I see that uses tokens from different token levels to conduct mixup operation and results in good performance in image classification.

- The analysis of this paper is great. Both the main paper and the supplementary material provide sufficient experiments to demonstrate how the proposed approach works. I really like the analysis and thank the authors for providing these experiments.

Despite the strengths in novelty and method analysis, I still have the following concerns.

Cons:

- Despite the efficiency, the proposed method needs an extra ScoreNet to compute saliency. This would inevitably introduce extra training time, no matter how much it is.

- In the original mixup method, there is only one hyper-parameter that needs to be tuned. However, in this paper, there are three new hyper-parameters as analyzed in the supplementary material. For CIFAR and ImageNet, the authors have provided suggestions on how to select them. However, when applied to other datasets, users still need to determine how to select them.

- At present, only CCT and the original ViT are used as baselines. Since the first ViT work, there are a lot of new ViT models. I really think the authors should conduct experiments on more powerful ViT models, like Swin Transformer, Pyramid Vision Transformer, and more recent VOLO, etc. Results on powerful baseline models would definitely make this paper stronger.

- In addition, from Table 5, we can see that the improvement is mostly from Input Mixup & Cutmix. Could the authors explain why the proposed approach succeeds while Manifold Mixup fails in lifting model performance?

---

> ### Author Response · Authors · 2022-08-02
> **Response for Reviewer pM5D**
>
> Thank you very much for the strong support for our method’s novelty. We are glad that the reviewer enjoyed our analyses. The questions will be addressed below.
>
> **Question 1. Is there extra time complexity in using ScoreNet?**
>
> The ScoreNet is a small MLP classifier, which is applied to an intermediate layer. Considering its size, ScoreNet adds close-to-zero overhead. When measured on CIFAR-100 with RTX 3090 using batch size of 128, the average latency caused purely by ScoreNet (both forward and back propagation) was 0.004 second, which is only 1.9% of the average latency of one iteration. This is a negligible scale.
>
> **Question 2. Need for exhaustive hyperparameter search?**
>
> We have answered the same question for reviewer Muyi. For your convenience, we repeated the answer here.
>
> Like most data augmentation methods, hyperparameter optimization is inevitable. However, our methods have at most two major hyperparameters; HTM entails  $\rho$ and $\tau$, while VTM entails $\kappa$. This is similar to previous Mixup methods; Input Mixup [1] (or Manifold Mixup [2]) with one hyperparameter and Puzzle Mix [3] with 4 major hyperparameters. Furthermore, as provided in the sensitivity analysis in Table 1 in Section C of the supplement, our method is quite robust to a wide range of hyperparameters.
>
> [1] Zhang, H., Cisse, M., Dauphin, Y. N., & Lopez-Paz, D. (2018). mixup: Beyond Empirical Risk Minimization. *ICLR*.
>
> [2] Verma, V., Lamb, A., Beckham, C., Najafi, A., Mitliagkas, I., Lopez-Paz, D., & Bengio, Y. (2019). Manifold mixup: Better representations by interpolating hidden states. *ICML*.
>
> [3] Kim, J. H., Choo, W., & Song, H. O. (2020). Puzzle mix: Exploiting saliency and local statistics for optimal mixup. *ICML*.
>
> **Question 3. Request for additional baseline models on ImageNet.**
>
> As suggested, we applied Horizontal TokenMixup with minor modification to the Pyramid Vision Transformer to evaluate its performance on ImageNet-1K. Please have a look at the general comment for experiment tables. We have also trained the ViT-Lite on CIFAR-100.
>
> However, PVT uses a convolution layer for key/value token projection. So, PVT is not resilient to the change in the number of tokens. Thus, we could not report the performance with VTM.
>
> **Question 4. Why does Manifold Mixup fail in contrast to TokenMixup?**
>
> In AlignMixup [1], LeViT was trained on Imagenet with Input Mixup, Cutmix and Manifold Mixup. Input Mixup and Cutmix achieved top-1 accuracy of 68.3% and 68.7%  each, while Manifold Mixup only achieved 67.8%. We believe this aligns with our observation that manifold mixup does not work relatively well in transformer models.
>
> Furthermore, we observed that Manifold Mixup severely damages the original attention map, becoming too smooth with frequent artifacts. Please have a look at Section J of the revised supplement for relevant qualitative examples; we have added Figure 3. We randomly sampled 8 mixed pairs each to compare the attention maps from the layer after TokenMixup and Manifold Mixup. Specifically, several attention heads (*e.g.* head 1 and head 2 in Supplement Figure 3) seemed to be defunct, rendering a very smooth attention distribution regardless of input. We believe these examples explain why Manifold Mixup did not perform well in our experiment settings. Note, the attention maps were retrieved from the CCT model trained on CIFAR-100.
>
> [1] Venkataramanan, S., Kijak, E., Amsaleg, L., & Avrithis, Y. (2022). AlignMixup: Improving Representations By Interpolating Aligned Features. CVPR.

---

### Official Review · Reviewer_NLuh · 2022-07-10

**Rating:** 5
**Confidence:** 2
**Soundness:** 3 good
**Presentation:** 3 good
**Contribution:** 3 good

**Summary:**

This paper presents a token-level data augmentation method that is tailored to networks that have attention layers. The authors argue that recent gradient-based mixup methods are costly and propose a horizontal (mixing tokens between different samples) and vertical (mixing tokens between different layers) mixup schemes using attention. Both proposed methods improve the performance of baseline models on CIFAR and ImageNet.

**Questions:**

- The authors provide results on HTM and VTM, but does not provide results when both HTM and VTM are applied simultaneously. How is the performance of HTM + VTM?
- Can the authors provide results on other transformer architectures to see if the proposed methods are generalizable?

**Limitations:**

The main limitation of the method is that it is only applicable to attention-based models, which the authors address clearly on the appendix.

**Strengths And Weaknesses:**

Strengths
- As attention-based architectures (e.g. transformer) are becoming more and more important in many deep learning domains, finding a mixup method tailored for those architectures is timely and valuable.
- The proposed method is much more cost-efficient compared to the previous saliency-based methods, which is especially important as transformers models are usually very large.

Weaknesses
- Since the proposed method consists of many components (Sample difficulty assessment, optimal assignment, token-level mixup, saliency-based label reassignment), I would like to see more ablations to verify if each component is necessary. For example, how does (sample difficulty assessment + Cutmix) work compared to HTM or VTM?
- The most direct comparison to the saliency-based mixup baselines are given in Table 3 and L228-233. The authors note that the proposed attention-based approach, while being much more cost-efficient, shows higher performance than the gradient-based methods such as Co-Mixup. However, the authors do not provide any intuition on why the attention-based approach is better. Therefore, it is hard to interpret the performance gap especially since the authors motivate their approach as an efficient alternative to the gradient-based methods.
- According to the results in Appendix C Table 1, the proposed method seems somewhat sensitive to the choice of hyperparameters. If one or two of the hyperparameters are not set properly, the performance drops similar to the baseline (CCT-7/3x1 + Input Mixup & Cutmix in main paper Table 5.).

---

> ### Author Response · Authors · 2022-08-02
> **Response for Reviewer NLuh**
>
> We appreciate the insightful comments and suggestions. We tried to address all the reviewer's questions below. We hope that our answers resolve the reviewer's concerns and lead to stronger support.
>
> **Question 1. Request for additional ablations for the TokenMixup components.**
>
> First of all, we provided a thorough ablation study of all the four components in Table 4 of the main paper.  As requested, we here provide additional ablation study combining with Input Mixup & Cutmix. Since all our modules, except for “Sample difficulty assessment”,  work with tokens and their attention scores, the only valid combination with image-level mixup methods is “Input Mixup & Cutmix + Sample difficulty assessment”.  The experimental results are presented in the table below. Interestingly, we observed that ScoreNet slightly improved other mixup methods, while falling short compared to our HTM and VTM.
>
> | Component compositions | CIFAR-100 Accuracy |
> |---|:--:|
> | CCT-7/3x1 + Input Mixup & Cutmix | 82.87 |
> |   &nbsp;&nbsp; &nbsp;&nbsp; (+) Sample difficulty assessment (ScoreNet) | 82.96 |
> |   &nbsp;&nbsp; &nbsp;&nbsp; (+) Horizontal TM | 83.55 |
> |   &nbsp;&nbsp; &nbsp;&nbsp; (+) Vertical TM | 83.42 |
>
> **Question 2. Why is the attention-based approach better than the gradient-based approach?**
>
> In our qualitative analysis, we observed that gradient-based saliency maps occasionally have irregular representation of the input image, while attention-based saliency maps are usually smoother than gradient maps. To quantitatively demonstrate this tendency, we here compare the smoothness statistics of the two saliency maps. In the table below, we provide the average Total Variation and Variance of the saliency map. Total variation is derived either with the L1 or L2 norm, each computed as
>
> $\begin{equation} TV_{L1} =  \sum_{i,j} |S_{i+1,j} - S_{i,j}| + |S_{i,j+1} - S_{i,j}|\end{equation}$
>
> $\begin{equation} TV_{L2} = \sum_{i,j} \sqrt{|S_{i+1,j} - S_{i,j}|^2 + |S_{i,j+1} - S_{i,j}|^2}\end{equation}$
>
> where $S$ is the attention or gradient-based saliency map. Also note that both saliency map was normalized to sum to 1. From the table above, we can see that the gradient-based method renders higher TV and variance, indicating that the overall distribution is sharper than attention maps.
>
> |  | Attention-based | Gradient-based |
> |--|:--:|:--:|
> |Total Variation (L1) | 0.5454 | 1.3176 |
> |Total Variation (L2) | 0.1441 | 0.3910 |
> |Variance | 4.88E-6 | 1.95E-5 |
>
> **Question 3. The method seems to be sensitive to hyperparameter settings.**
>
> We have shown that our method is robust to the choice of hyperparameters, in section C of the supplement. The performances were consistently above their baselines.
>
> As a concrete comparison with previous works, Puzzle Mix, which was published in ICML 2020, reported the standard deviation for CIFAR-100 errors across reasonable hyperparameter ranges. For hyperparameters $\beta, \gamma, \eta, \xi$, the reported standard deviations were 0.22, 0.20, 0.18, 0.27, respectively. In HTM, the standard deviations of $\rho, \tau$ are 0.24 and 0.29, while $\kappa$ in VTM is 0.20. Considering that our sensitivity tests also included the harshest settings, we believe that our method is not especially sensitive to hyperparameters compared to previous outstanding works.
>
> | PuzzleMix parameters | Range | Mean Top-1 Error % (STD) |
> |:--:|--|:--:|
> |$\beta$| [0.8, 1.6] | 16.19 (0.22) |
> |$\gamma$| [0.0, 1.0] |  16.43 (0.20) |
> |$\eta$| [0.1, 0.35] |  16.37 (0.18) |
> |$\xi$| [0.4, 1.0] | 16.25 (0.27) |
>
>
> | TokenMixup parameters | Range | Mean Top-1 Error % (STD) |
> |:--:|--|:--:|
> |$\tau$| {0.0, 1.0, 1.5, 2.0, 2.5, inf} | 16.84 (0.24) |
> |$\rho$| {0.0, 0.001, 0.003, 0.005, 0.007, 0.010, inf} | 16.83 (0.29) |
> |$\kappa$| {0, 3, 5, 10, 25, 50, 100} | 16.89 (0.20) |
>
> **Question 4. How is the performance of using HTM and VTM simultaneously?**
>
> We report experimental results with CIFAR-100 models, CCT and ViT-Lite, when Horizontal TokenMixup and Vertical TokenMixup are applied simultaneously. From the table below, we can observe that utilizing both methods consistently improves its baseline performance as well as the performance when only one of the methods is used.
>
> | | ViT-Lite-7/4 | CCT-7/3x1 |
> |--|:--:|:--:|
> | vanilla model | 79.44 | 82.87 |
> | &nbsp;&nbsp;&nbsp;&nbsp; (+) Horizontal TM (ours) | 80.53 | 83.55 |
> | &nbsp;&nbsp;&nbsp;&nbsp; (+) Vertical TM (ours) | 80.44 | 83.42 |
> | &nbsp;&nbsp;&nbsp;&nbsp; (+) Horizontal TM + Vertical TM (ours) | **80.65** | **83.56** |
>
> **Question 5. Request for an experiment on other transformer architectures.**
>
> We further applied our methods to two more transformer architectures: ViT-Lite-7/4 with CIFAR-100 and Pyramid Vision Transformer-B0 with Imagenet-1K. Please refer to the general comment above for experimental tables. We observed that our methods consistently improve their baselines.

---

> > ### Comment · Reviewer_NLuh · 2022-08-07
> > **Response for the authors**
> >
> > Thanks for the detailed clarifications and additional experiments.
> >
> > The response addresses most of my concerns. Especially, the response to Q2 seems to show an interesting difference between gradient-based methods and attention-based methods. Also, the response to Q3 seems fair enough.
> >
> > Based on the response, I have raised my score.

---

### Official Review · Reviewer_nzAU · 2022-07-11

**Rating:** 7
**Confidence:** 4
**Soundness:** 3 good
**Presentation:** 4 excellent
**Contribution:** 4 excellent

**Summary:**

In this paper, the authors proposed a novel saliency-aware token-level data augmentation method for assisting the training of vision transformers. Specifically, they proposed TokenMixup, which takes the self-attention as the inherent saliency detector, to mix intermediate token sets so that the saliency level of a batch is maximized. With TokenMixup, they achieve significant improvement on image classification tasks and meanwhile obtain higher efficiency than other gradient-based mixup approaches.

**Questions:**

1. Is there any other methods for replacing the ScoreNet for filtering data samples with higher confidence?

**Limitations:**

There is no potential negative societal impact for this work.

**Strengths And Weaknesses:**

Pros:

1. This paper proposes TokenMixup, a novel saliency-aware token-level mixup strategy for improving vision transformers. It ingeniously adopts self-attention, the basic module in transformers, to serve as a substitute for conventional computationally-heavy saliency detector.

2. The authors conduct thorough experiments for evaluating the effectiveness of the proposed method by comparing different network backbones and different augmentation approaches. The ablation study is sufficient for validating the importance of each component of TokenMixup as well.

3. Based on the proposed TokenMixup, the authors propose a variant: vertical TokenMixup. By incorporating the features from previous layers, it can easily captures multi-scale feature for further improvement, which is simple and effective.

4. The authors provide some explanations and discussions from a perspective of curriculum learning, which leaves some inspirations for further research.

Cons:

1. Since ScoreNet is trained simultaneously with the main branch, there may exist some biases during the calculation of overall saliency.

---

> ### Author Response · Authors · 2022-08-02
> **Response for Reviewer nzAU**
>
> Thank you for the detailed review and comments. We appreciate your strong support for our work. The questions will be addressed below.
>
> **Question 1. Does ScoreNet cause bias in saliency estimation if trained simultaneously with the main model?**
>
> This is a very interesting question. We have investigated the effect of the supervision on ScoreNet by applying stop-gradient before ScoreNet propagation. However, we did not find a huge difference in the experimental results, implying that the estimated saliency map is not severely perturbed by supervision signals from ScoreNet training. We concluded that whether to use stop-gradient can be empirically determined. Specifically, we applied stop-gradient in our Imagenet experiment and did not in the CIFAR experiments.
>
> **Question 2. Are there methods other than ScoreNet for filtering data samples with higher confidence?**
>
> ScoreNet is a small classifier applied to an intermediate layer. A straightforward alternative would be to directly use the final confidence score as a proxy of sample difficulty. Relevant experimental results were reported in Section F of the supplement. In the experiment, we found that the method performed inferior to ScoreNet, while even slowing down iteration time by 47% due to an additional forward pass to compute the final confidence score.
>
> Other possible options would be uncertainty (*e.g.* accepting the difficulty score only when entropy is low). However, computing uncertainty may require Monte Carlo sampling or complex model structures. Thus, exploring other alternatives is an interesting future direction but we believe that our ScoreNet is an efficient and effective method to filter difficult samples.

---

### Official Review · Reviewer_Muyi · 2022-07-13

**Rating:** 7
**Confidence:** 3
**Soundness:** 3 good
**Presentation:** 3 good
**Contribution:** 3 good

**Summary:**

The paper proposes a novel data augmentation technique for visual transformers.

**Questions:**

- Can you encapsulate your staff in a layer and show how to add it? The source code is based on a function. If you want your code to be used by others, having it as a standard module and a simple way to incorporate it in existing models would be a great plus to your work

**Ethics Review Area:**

["I don’t know"]

**Strengths And Weaknesses:**

Strengths:

- Interesting and novel idea
- Faster performance
- SOTA results
- Comprehensive ablation studies

Weaknesses:

- It is not entirely clear what does it take and how easy it is to add this method to an arbitrary transformer. It would help to have an appendix with python code showing how a typical transformer is modified with the proposed algorithm. Is it as simple as adding a layer for which good hyperparameter values are known, or additional hyperparameter search is really necessary on case by case basis?
- The work explores 2 baseline transformer methods applied to 3 datasets. There are other models that are cited in the paper such as ViT-Lite, NesT-T, NesT-B. It would be of interest to have your work applied to those as well.
- I feel this method could have wider applicability, beyond Transformer based models (e.g. ResNets). It would be of interest to either see some empirical results exploring this area, or if this is impossible due to the limitations of the method, to have a discussion of this limitation and perhaps an outline of how this limitation could be overcome in future work.

---

> ### Author Response · Authors · 2022-08-02
> **Response for Reviewer Muyi**
>
> Thank you very much for the comments. We especially appreciate the suggestions regarding code release. Below are our responses to the reviewer's questions.
>
> **Question 1. What does it take to add TokenMixup to Transformers, and is hyperparameter search necessary?**
>
> Based on the code we provided in the supplement, several alterations may be required. For instance, if the original Transformer module does not save the token and attention map (*i.e.* the saliency map) for the self-attention layers, additional lines of code will be needed to utilize our TokenMixup functions. However, architectural change was not needed, and therefore can be applied with ease. As requested, we provide the PyTorch-style pseudo-code below which describes how our method is utilized overall. Moreover, as suggested, we are planning to improve code readability & usability and modularize our source code for wider use, on acceptance.
>
> Like most data augmentation methods, hyperparameter optimization is inevitable. However, our methods have at most two major hyperparameters; HTM entails  $\rho$ and $\tau$, while VTM entails $\kappa$. This is similar to previous Mixup methods; Input Mixup [1] (or Manifold Mixup [2]) with one hyperparameter and Puzzle Mix [3] with 4 major hyperparameters. Furthermore, as provided in the sensitivity analysis in Table 1 in Section C of the supplement, our method is quite robust to a wide range of hyperparameters.
>
> ```
> class TypicalTransformer :
> 	def __init__(self, **kwargs):
> 		self.layers = self.build_layers(kwargs)
> 		self.mixup_type = kwargs['tokenmixup_type']
> 		self.applied_layer = kwargs['applied_layer']
>
> 	def build_layers(self, **kwargs) :
> 		"""
> 		    code for initialization of transformer layers
> 		"""
> 		return layers
>
> 	def token_mixup(self, **kwargs):
> 		"""
> 		    Horizontal or Vertical TokenMixup code
> 		"""
> 		return x, y
>
> 	def forward(self, x, y):
> 		attention_maps = list()
> 		previous_tokens = list()
> 	        for i, layer in enumerate(self.layers) :
> 		    if self.applied_at == i and self.mixup_type == 'horizontal' :
> 		        x, y = self.token_mixup(x, y)
> 		    elif self.applied_at == i and self.mixup_type == 'vertical' :
> 		        x, y = self.token_mixup(x, y, previous_tokens, attention_maps)
> 		    previous_tokens.append(x)
> 		    x, attention = layer(x)
> 		    attention_maps.append(attention)
> 		return x, y
> ```
>
> [1] Zhang, H., Cisse, M., Dauphin, Y. N., & Lopez-Paz, D. (2018). mixup: Beyond Empirical Risk Minimization. *ICLR*.
>
> [2] Verma, V., Lamb, A., Beckham, C., Najafi, A., Mitliagkas, I., Lopez-Paz, D., & Bengio, Y. (2019). Manifold mixup: Better representations by interpolating hidden states. *ICML*.
>
> [3] Kim, J. H., Choo, W., & Song, H. O. (2020). Puzzle mix: Exploiting saliency and local statistics for optimal mixup. *ICML*.
>
>
> **Question 2. Request for additional experiments on CIFAR baseline models.**
>
> We further applied our methods to ViT-Lite with CIFAR-100, as well as on Pyramid Vision Transformer with Imagenet-1K. The experimental results are presented in the general comment above. In the experiments, we observed consistent improvements in performance compared to the baselines.
>
>
> **Question 3. Can TokenMixup be applied beyond Transformer based models?**
>
> The major target of our framework is parameter-heavy transformers with explicit attention maps. Hence, our framework was not built for non-transformer models without self-attention layers. Such limitation was discussed in section H of the supplement, and that Horizontal TokenMixup would be applicable if additional attention modules are adopted. On the other hand, Vertical TokenMixup-like feature map augmentation for convolution layers was previously not dealt with in literature. Although it is yet not straightforward to directly concatenate multi-scale features without affecting spatial structure, it would be an interesting research direction to augment feature maps in a vertical manner.

---

### Author Response · Authors · 2022-08-02
**General Response for ALL reviewers**

We thank all four reviewers for their strong support and constructive comments on our work. We are glad that the reviewers found our method novel and interesting. Here, we added a few more experimental results on CIFAR-100 and Imagenet-1K.

As suggested by the reviewers, we applied our method to one of our baselines, ViT-Lite-7/4, for the CIFAR-100 experiment. In the table below, * denotes the reproduced baseline performance by training for longer (1500) epochs. In the case of Imagenet experiments, we applied Horizontal TokenMixup to the PVTv2-B0 model. We observed meaningful improvements in performance in both settings.

|      | CIFAR-100 Accuracy |
|-----|:-----------------:|
| VIT-Lite-7/4 (1500) * | 79.44 |
| VIT-Lite-7/4 (1500) + **Horizontal TM (ours)** | **80.53** |
| VIT-Lite-7/4 (1500) + **Vertical TM (ours)** | 80.44 |


|      | ImageNet-1K Top1 | ImageNet-1K Top5 |
|-----|:-----------------:|:-----------------:|
| PVTv2-B0 | 70.46 | 90.16 |
| PVTv2-B0 + **Horizontal TM (ours)** | **71.20** | **90.43** |

---

### Author Response · Authors · 2022-08-07
**A Gentle Reminder for Reviewers**

Dear reviewers,

We appreciate the insightful comments and your time in reviewing our paper. We have responded to the reviewers' questions and uploaded the revised version of the supplement. Please go over our responses and let us know if you have any further questions. Thank you!

---

### Meta-Review · Area_Chair_kGPv · 2022-08-25

**Recommendation:** Accept
**Confidence:** Certain

**Metareview:**

The paper is about speeding up the saliency computation used in gradient based mixup algorithms(Puzzlemix, Co-mixup, etc). The authors propose employing the attention layer output of the transformer to replace the expensive saliency computation.

Since focus of the the paper is on improving the speed and accuracy aspects of gradient based mixup algorithms, the tables 1 and 2 should remove

I strongly suggest the authors to remove tables 1 and 2 as it only shows irrelevant accuracy results across different network architectures. Since the main focus of the paper is on improving over the previous mixup algorithms, I suggest the authors to run the following experiment and insert it as table 1 in the main paper. Current table 5 does not show fair comparison across different mixup methods.

1) fix the network architecture. Denote it as A (i.e. CCT-7/3x1)
2) report the following accuracy results.
A;
A + input mixup;
A + manifold mixup;
A + cutmix;
A + puzzlemix;
A + co-mixup;
A + horizontal TM;
A + vertical TM;
A + horizontal TM + vertical TM

Here, you would need to make sure the backpropagate all the way through the individual pixels as opposed to tokens to properly run the gradient based mixup algorithms.

Also, I suggest the authors to add the end-to-end forward-backward computation time per image in addition to the avg. latency in table 3.

Overall, I like the proposed method of speeding up the saliency computation via attention maps. However, the experiment protocol needs a lot of improvement.

**Award:**

No

---

### Decision · Program_Chairs · 2022-09-14

Accept